# Accumulation and Enrichment of Trace Elements by Yeast Cells and Their Applications: A Critical Review

**DOI:** 10.3390/microorganisms10091746

**Published:** 2022-08-30

**Authors:** Jie Sun, Shiyi Xu, Yongbao Du, Kechen Yu, Yi Jiang, Hao Weng, Wei Yuan

**Affiliations:** 1Key Laboratory of Bioorganic Synthesis of Zhejiang Province, College of Biotechnology and Bioengineering, Zhejiang University of Technology, Hangzhou 310014, China; 2Hangzhou Se-easy Biotechnology Co., Ltd., Hangzhou 311100, China

**Keywords:** *Saccharomyces cerevisiae* (*S. cerevisiae*), trace elements, chromium-, iron-, zinc- and selenium-enriched yeast, metabolic engineering

## Abstract

Maintaining the homeostasis balance of trace elements is crucial for the health of organisms. Human health is threatened by diseases caused by a lack of trace elements. *Saccharomyces cerevisiae* has a wide and close relationship with human daily life and industrial applications. It can not only be used as fermentation products and single-cell proteins, but also as a trace elements supplement that is widely used in food, feed, and medicine. Trace-element-enriched yeast, viz., chromium-, iron-, zinc-, and selenium-enriched yeast, as an impactful microelements supplement, is more efficient, more environmentally friendly, and safer than its inorganic and organic counterparts. Over the last few decades, genetic engineering has been developing large-scaled genetic re-design and reconstruction in yeast. It is hoped that engineered yeast will include a higher concentration of trace elements. In this review, we compare the common supplement forms of several key trace elements. The mechanisms of detoxification and transport of trace elements in yeast are also reviewed thoroughly. Moreover, genes involved in the transport and detoxification of trace elements are summarized. A feasible way of metabolic engineering transformation of *S. cerevisiae* to produce trace-element-enriched yeast is examined. In addition, the economy, safety, and environmental protection of the engineered yeast are explored, and the future research direction of yeast enriched in trace elements is discussed.

## 1. Introduction to Trace Elements

Micronutrients play an important role in maintaining the health of living things. The regulation of micronutrients on body metabolism is an important research topic for nutrition science and also an important biological research area. An expert committee of the World Health Organization (WHO) classifies the essential trace elements into three categories [1] which include essential trace elements: iodine (I), iron (Fe), zinc (Zn), selenium (Se), copper (Cu), molybdenum (Mo), chromium (Cr), and cobalt (Co). The second class includes manganese (Mn), silicon (Si), nickel (Ni), boron (B), and vanadium (V). The third class is the trace elements which are potentially toxic but may have necessary functions for the human body at low doses, including fluorine (F), lead (Pb), cadmium (Cd), mercury (Hg), arsenic (As), aluminum (Al), lithium (Li), and tin (Sn).

These trace elements play significant roles in biological systems, which are involved in gene regulation, nucleic acid metabolism, anti-inflammatory and antioxidant functions, and other special physiological functions. Different trace elements play different physiological functions in the body. Chromium functions in maintaining the normal metabolism of glucose, lipid, and protein in the body. It can maintain the dynamic balance of blood glucose and improves levels of blood sugar and lipid to a certain extent [2,3]. Chromium is also an important active component of glucose tolerance factor (GTF), which increases the sensitivity of tissue receptors towards insulin [4]. GTF can regulate the metabolism of biological macromolecules, deposition of muscle tissue, and utilization of cholesterol via increasing the affinity between the tissue receptors and insulin [5]. Meanwhile, chromium is also considered to be a promising anti-heat stress element that can reduce the use of antibiotics because of its strong antioxidant effect, which prevents reactive oxygen species (ROS) from degrading lipid film structures, resulting in lipid peroxidation and cell damage [6].

A grain-based diet generally causes iron deficiency. Iron deficiency leads to anemia because iron is a component of hemoglobin (a blood protein) [7]. Iron is involved in transporting oxygen through red blood cells, so iron deficiency may exacerbate chronic inflammation [8]. Iron also participates in many essential biochemical processes, such as the synthesis of deoxyribonucleotides, amino acids, lipids, and sterols. As a component of cytochrome (CYP) enzymes, iron participates in the electron transport chain and is pivotal to oxidative phosphorylation and redox reactions involved in the respiratory chain [9].

The element zinc contributes to cell growth and apoptosis. Zinc is abundant in the brains of mammals, and insufficient intake of zinc can affect intellectual development and immunity among children [10]. Zinc is also important for the central nervous system [11]. Zinc deficiency activates inflammatory responses which trigger ROS-induced oxidative stress [12]. Copper stimulates the growth of livestock and poultry, so it is often added to animal feed [13]. The molybdenum cofactor, which is required for purine metabolism and sulfite detoxification, is synthesized with molybdenum [14].

Selenium is a component of selenocysteine and an essential component of selenoproteins and selenases such as glutathione peroxidase (GPx), phospholipid hydrogen glutathione peroxidase (PHGPx) and thioredoxin reductase (TrxR) [15]. Approximately 10 additional selenoproteins have been identified. Selenium in the form of selenocysteine is the active center of many antioxidant enzymes in the body. Selenium is related to human antioxidant activity and anti-inflammatory and anti-virus properties [16]. Selenium deficiency can induce or increase skeletal muscle and myocardial necrosis, immune function decline, and tumor diseases [17].These micronutrients are important for health, but excessive consumption is toxic [18]. A high level of metal trace elements may cause inflammation in the body, induce an oxidative stress response, and may interfere with the absorption of other trace elements [12]. A large amount of selenium will cause acute toxicity, leading to diarrhea, vomiting, and other symptoms. Serious overdoses can cause heart problems or even death [19]. Continuous intake of selenium slightly higher than the toxic dose may cause chronic poisoning. It will cause skin rash, hair loss, etc., and also have a negative impact on the digestive system and nervous system [20]. Specific, adequate, and standard intakes of essential micronutrients are required to minimize the risk of nutrient deficiency or their excess. At the same time, the toxicity of metal trace elements in different valence states is different. Additionally, there is antagonism between metals, i.e., the increase in the absorption of one metal could result in the decrease of the absorption of another metal. Therefore, an appropriate trace element supplement is very important to maintain the balance of trace elements in the body [21].

## 2. Trace Element Supplement Form

Humans are not able to meet the body’s nutritional demand for trace elements from food alone, which leads to the steady state of trace elements in the human body often tending to be insufficient. For example, the estimated safe and adequate daily dietary intake (ESADDI) of human chromium is 50–200 μg [22]. It is hard to obtain this amount through daily meals. Moreover, about 1 billion people in the world have insufficient selenium intake, covering both developed and developing countries [23]. If trace elements are consumed in excessively low amounts, they can adversely affect various organs, including the immune system, the nervous system, and the metabolic system. Micronutrient deficiencies cause severe diseases such as iron deficiency anemia, night blindness, and even cancer [24,25]. Micronutrients are also found to be deficient in varying degrees at different ages in the body, and specifically, children have higher micronutrient requirements in comparison to adults [26]. At present, more than 2 billion people suffer from micronutrient deficiency [27]. It can be seen that it is far from enough to supplement trace elements merely from food. It is not surprising that mineral element deficiency is a common problem in both developing and underdeveloped nations.

### 2.1. Inorganic Compounds

There are currently mainly two types of chelating supplements available on the market: inorganic and organic. Trace elements in inorganic form are mainly inorganic salts whose production methods are simple. They are easy to produce on a large scale, but the human body usually has a slow digestion and utilization rate of inorganic salt. The inorganic salt forms of many trace elements do not possess biological activity, so they need to be further transformed into biologically active chemical forms by various physiological and biochemical processes. In the meantime, inorganic trace elements are usually toxic, and their improper use may cause severe adverse effects. For example, inorganic forms of chromium supplements include chromium chloride, chromium picolinate, and other forms of trivalent chromium [28]. Trivalent chromium, though, is less toxic than other forms of chromium. The only adverse effects found in humans with improper consumption of chromium salts in very high doses have been linked to liver and kidney health [29]. This is due to the difficulty in absorbing and utilizing inorganic chromium in the gastrointestinal tract of mammals. Indigestible inorganic chromium can accumulate in the kidneys and liver. It was found that the chromium content in the kidney after consuming inorganic chromium for 24 h was 10 times that of taking inorganic chromium for 1 h, and it was 5 times that of taking inorganic chromium in the liver. This situation can last up to six months [30]. Parenteral supplementation of zinc is usually made with zinc chloride or zinc sulfate. Similar to inorganic chromium, inorganic zinc is also not conducive to absorption by the human body, and it exhibits the same characteristics as inorganic chromium of low absorption efficiency. In addition, inorganic zinc is unstable and tends to deteriorate during production, transportation, and storage. Inorganic forms of selenium include selenate (Na_2_SeO_4_) and sodium selenite (Na_2_SeO_3_) [31], but inorganic selenium may cause genotoxicity in the human body [17]. Due to the low digestibility of inorganic forms of trace elements, to meet the human body’s demand, it is necessary to increase the dose, which not only aggravates the toxicity and side effects on human health, but also causes an environmental burden. For example, about 80% of the supplemented inorganic copper is discharged through feces, adding pollution to the natural ecosystem [32].

### 2.2. Organic Chelates

The organic form of trace elements is usually salts of organic acids or organic chelates. Compared with inorganic forms, it is less toxic and better utilized by organisms. However, due to the different organic forms, it is impossible to obtain multiple organic trace elements at the same time. The safety of synthetic organic compounds is also controversial. As food and feed additives, iron chelates (ferrous diglycine) are less sensitive to iron absorption inhibitors than iron salts. Zinc supplements can be taken orally in the form of salts of organic acids, such as zinc gluconate, zinc acetate, or zinc propionate. The complementary form of organic copper is mainly the complex of copper in combination with gluconic acid. Organic chromium as a trace element supplement includes phenylalanine chromium, nicotinate chromium, GTF-like chromium pyridinate, and so on [33]. Organic chromium has better performance both in terms of rate of absorption and bioactivity. Chromium picolinate, in particular, is one of the most popular forms of chromium supplements and has been marketed as a promoter of fat loss, muscle gain, and treatment for metabolic disorders including type II diabetes by reducing blood glucose levels on the premise of increasing insulin [34]. Nevertheless, organic chromium, especially chromium picolinate, has been reported to be harmful to humans and may cause cancer [35]. Organic chromium compounds are indeed more easily absorbed than inorganic chromium compounds, but even so, the absorption rate is seen to be <1% [36]. The organic supplementary forms of selenium include selenomethionine (SeMet) and selenocysteine (SeCys) [37]. Selenium exists in humans in the form of SeMet. As a result, SeMet is easier to be absorbed by the body than the inorganic form [38]. In summary, inorganic compounds generally are more toxic and indigestible than organic ones.

### 2.3. Yeast Cells Enriched in Trace Elements

*S. cerevisiae* is generally recognized as a safe (GRAS) microbe. Humans have used yeast to make fermented products for thousands of years. Nowadays, as a common food additive and nutritional supplement, yeast has a closer relationship with human life. The biomass of *S. cerevisiae* is especially rich in protein and contains eight essential amino acids and is expected to become an important source of protein in the future [39]. The protein content of its biomass can reach to 54% [40]. Because of its rich protein content, it can better enrich iron, zinc, chromium, selenium and other trace elements. Yeast biomass is used as a feed or food supplement. Because of its high bioavailability, it is mainly used as a source of micronutrient supplements.

As compared to the microelement preparations being synthesized by a chemical method, the microelements from yeast have higher biological activities and can be absorbed easily in the body. Moreover, the fermentation process of yeast is a simple and short operation. This makes yeast an ideal carrier for the production of metal trace element preparations. At present, industrial trace-element-enriched yeast is mainly produced by adding mineral salts of trace elements to the fermentation medium to complete the enrichment of trace elements in the metabolic process of *S. cerevisiae*. Screening and mutagenesis of yeast with high enrichment [41] and optimization of fermentation conditions [42] are the traditional ways to improve the enrichment of trace elements in yeast. The selection of inorganic salts of trace elements in the culture medium possibly impacts the enrichment of *S. cerevisiae*. For example, trivalent chromium shows quite low toxicity in several valence states of chromium [43]. Choosing chromium trichloride as a substrate might reduce the detoxification burden of *S. cerevisiae*.

In recent years, microorganisms, especially yeast, have been utilized as carriers for trace elements to obtain organic trace elements that are easier to digest and absorb with enhanced biological activity. They are also inexpensive to manufacture. In addition to the low cost, trace-element-enriched yeast can effectively replace other inorganic and organic trace element nutritional supplements. Moreover, yeast can also be used for soil treatment, water purification, and in many other fields. Based on the above advantages and application prospects in multiple fields, yeast as a source of trace elements has attracted intensive attention from the research and market point of view.

#### 2.3.1. Chromium Yeast

The active form of chromium in yeast is GTF, which is formed by the coordination of trivalent chromium ions with amino acids. Yeast containing chromium is considered an ideal microelement supplement [44]. A study by Krol et al. [45] found that compared with chromium chloride, the addition of chromium-enriched yeast during chicken feeding could reduce the content of crude fat and dry matter in chicken chest and leg muscles. Another study carried out by Liu et al. [28] compared the bioactivity of different chromium-based compounds by using insulin-resistant 3T3-L1 adipocytes and showed that GTF can improve glucose metabolism much more efficiently than other forms of chromium such as chromium pyridinate or chromium trichloride.

At present, chromium-enriched yeast is mainly produced by yeast fermentation in the presence of chromium chloride [46]. Previous studies were mainly focused on optimizing fermentation conditions for improving chromium content and yield of chromium-enriched yeast. Ali et al. [41] carried out fermentation research on wild beer yeast, using a compound medium in batch-fed fermentation and sodium chromate as the chromium source. After 50 h of fermentation when the sodium chromate supplemental level was 7.1 g/L, the process yielded about 4.2 g/100 mL of chrome-enriched yeast biomass with a total chromium content of 3113 μg/g, where the organic chromium content was 795 μg/g.

#### 2.3.2. Zinc Yeast

Zinc-enriched yeast is a kind of common organic zinc source which is an ideal zinc additive. Zinc-enriched yeast can organically combine zinc with proteins and polysaccharide yeast through the absorption and transformation of zinc during yeast growth, thus eliminating toxic side effects and gastrointestinal irritation caused by inorganic zinc and organic zinc in the human body and making it possible for zinc to be absorbed and utilized by the human body more efficiently and safely. It is superior to other types of zinc in digestion and absorption as well as physiological transformation efficiency. Maares et al. [47] showed that zinc-enriched yeast is a promising nutritional supplement by comparing it with other zinc preparations such as zinc sulfate and zinc gluconate. Fan et al. [48] used siderophore isolated from an iron-rich environment to produce zinc–copper enriched yeast, and finally obtained *S. cerevisiae* with intracellular organic Cu and Zn contents of 60.76 and 44.22 mg/g, respectively.

#### 2.3.3. Iron Yeast

Iron supplements are usually associated with certain side effects and risks. They may cause gastrointestinal problems such as vomiting in up to 40% of patients [49]. In addition to their low absorption, these preparations are also associated with a metallic taste. Iron-enriched yeast can provide a better form of iron supplementation by being stable in form, absorbing easily, producing foods in coordination with other ingredients, and also having a good flavor. In a study conducted by Sabatier et al. [50], it was found that iron-enriched yeast exists in the form of a low molecular weight organic complex through HPLC-ICP-MS preliminary analysis. Nowosad et al. [51] used iron-enriched yeast to yield 100 g dry weight flatbread containing almost 385.8 ± 4.12 mg iron. Furthermore, this product had no metallic taste and can be used as a daily iron supplement. The study of Raguzzi et al. [52] showed that the yeast cells can store iron in two forms: one is stored in the form of cytoplasmic molecules similar to ferritin and the second is in vacuoles by binding iron with polyphosphates. By using interspecific protoplast technology and fermentation under optimal conditions, Yuan et al. [53] obtained iron-enriched yeast containing Fe at a concentration of 25 mg per gram of stem cells.

#### 2.3.4. Selenium Yeast

Selenium yeast is a successful case of *S. cerevisiae* enriching trace elements. Selenium-enriched yeast is widely used not only as a selenium nutritional supplement, but also in the food field to produce selenium-enriched milk, selenium-enriched chicken, and so on. Organisms can biotransform inorganic selenium and combine selenium with amino acids, proteins, polysaccharides and other substances to convert it into organic selenium. Selenium (SeMet, etc.) transformed by *S. cerevisiae* can be better absorbed by the human body [17]. At the same time, it shows less toxicity [54]. However, the research not only focuses on the selenium enrichment of yeast, but also derives a new form of selenium supplement (nano-selenium). Compared with selenium compounds, nano-selenium produced by *S. cerevisiae* has better biocompatibility, lower cytotoxicity, and better biological activity than inorganic selenium and organic selenium [55]. The production of nano-selenium depends on the reduction of glutathione in *S. cerevisiae* [56]. The selenium content of selenium-enriched yeast can reach 5.64 mg per gram [57]. Khoei et al. [58] treated sodium selenite with *S. cerevisiae* to rapidly reduce to nano-selenium. The analysis showed that the size of nano-selenium was 100–300 nm. In addition to common yeasts enriched in chromium, iron, zinc, and selenium, there is little chance that the human body will become deficient in other metal trace elements, and daily diets can meet the body’s nutritional requirements. Therefore, yeast nutritional supplement products with other metal trace elements are rare. Additionally, copper yeast and molybdenum yeast have been studied as yeast nutritional supplements. By screening high tolerance strains and optimizing the composition of the culture medium and fermentation conditions, Guo et al. [59] made it possible to increase the copper ion absorption rate of *S. cerevisiae* to 90%. In comparison with organic copper salt, copper-enriched yeast demonstrated a higher utilization rate [60].

## 3. Mechanism of Trace Element Transport in Yeast

The heavy metal binding in yeast can be achieved in two different ways: bio-adsorption and bio-enrichment. Bio-adsorption is a metabolically passive physiochemical process. Through electrostatic force, chemical bond, and other physical and chemical forces, metal and yeast extracellular polymers are combined, so that the metal particles adsorb to the surface of microorganisms. The anionic groups on the outer surface of yeast cells provide binding sites for positively charged heavy metals. Metal cations can bind to microbial surface components [61]. The composition of yeast cell walls determines its ability to adsorb metals. Chitin and glucan-mannoprotein complex are the main active substances in yeast adsorption [62].

Bio-enrichment is a metabolically dependent process that occurs in yeast cells. A number of basic metal transport components, including membrane transport proteins, organelle storage systems, and chelating agent molecules, ensure metal uptake and their storage by yeast. Heavy metals can be transported into yeast cells, which depend on the existence of phospholipid bilayers. In the outer phospholipid layer, heavy metals pass through pore proteins. Then they enter yeast cells through channels, secondary carriers, or primary active transporters. Numerous channels and transporters located on the cell membrane have specificity and can specifically recognize and transport different heavy metals [63]. It is an important way for yeast to transport heavy metals. Once inside the intracellular space, heavy metals can be bound by proteins and peptide ligands (glutathione, metallothionein, phytochelatin, etc.) to limit cytotoxicity and isolate the binding within the cell, thereby eliminating metals from sensitive metabolic functions [64,65]. To achieve metal enrichment, some metal cations bind to functional groups on the cell wall and are then internalized into the cell [66]. 

Yeast enriches metal trace elements by using both common and specific pathways. Several genes are involved in both pathways, including those involved in metal transporters, glutathione synthesis-related enzymes, and genes related to the synthesis of other metal-binding proteins. There are some genes that regulate the transport and enrichment of many metals, while others are only related to a single/specific metal. With current genetic engineering technologies, yeast can now be engineered on all levels, from specific proteins to complex metabolic pathways.

### 3.1. Chromium and Selenium Transport

Yeast transports chromium in two ways, one by diffusion or phagocytosis [67], which transports chromium ions directly into cells. The other method involves the transport of chromium metal via transporters located on cell membranes. Studies have shown that the transporter transports chromium that adsorbed on the cell wall into yeast cells [68]. Two high-affinity sulfate transporters, Sul1 and Sul2, exist in *S. cerevisiae*, and their expression is thought to be regulated by at least two transcriptional activators, MET4 and MSN1 [69]. This observation suggests that chromates enter the cells through the sulfate assimilation pathway [70].

The uptake of selenium by cells is mainly through the transport system of sulfate and phosphate. Selenite resistance is closely related to the expression of high affinity orthophosphate vector pho84p [71]. The molecular structure and spatial arrangement are similar for sulfate (SO_4_^2−^), chromate (CrO_4_^2−^), and selenate(SeO_4_^2−^) ions [72]. Therefore, Sul1 and Sul2 can also act as selenate transport channels.

### 3.2. Iron Transport

Yeast cells absorb iron mainly through enzymes and proteins on their cell membranes. The cellular systems involved in iron uptake and utilization are precisely regulated according to iron availability and the cellular requirement of iron. Iron depletion induces the expression of a family of cell wallproteins known as Fit1p, Fit2p, and Fit3p [73]. The yeast cells have evolved two iron transport systems to uptake iron at different concentrations. This is performed without causing iron toxicity in the yeast cells nor causing deficiency. Furthermore, yeast cells have evolved two iron transport mechanisms with a low-affinity [74] and a high-affinity system [75], which coordinate with the identification and absorption of iron.

These systems are responsible for iron transport in iron-sufficient and iron-deficient cells, respectively. Based on the analysis of yeast mutants, some genes related to iron transport have been cloned by specific mutations and effective screening. For example, plasma membrane metal reductases encoded by FRE1 and FRE2 genes can reduce ferric (Fe^3+^) iron [76]. Fe^2+^ transporter genes FTR1 and FET4 are located on the plasma membrane of yeast [77]. In *S. cerevisiae*, transcription factors Aft1/Aft2 and Yap5 regulate the metabolism of iron in response to low and high iron levels, respectively [78]. Lucia et al. [79] express constitutively active Aft1 alleles to increase the accumulation of iron in *S. cerevisiae*.

### 3.3. Zinc Transport

Similar to the transport of iron through yeast, concentration-dependent analysis of zinc uptake by yeast cells showed that there were at least two uptake systems which are involved: a high-affinity system in zinc-deficient cells [80] and a low-affinity transport system mainly expressed in zinc-abundant cells [81]. Zrt1 is a zinc transporter in the high-affinity transport system. When zinc is depleted in the cells, due to their high-affinity transport system and specificity for zinc, they can improve the absorption of zinc through absorption mediated by yeast [82]. The gene responsible for low-affinity zinc transport is Zrt2 [81]. Zap1 is a major regulator of zinc deficiency in *S. cerevisiae*, which regulates the expression of Zrt1 and Zrt2 [83]. Zinc is transported in vacuoles by two zinc transporters, Zrt3 and Zrc1. The transport of zinc is also regulated by the transcription factor Zap1 [84]. The mitochondrial carrier gene MTM1 can maintain zinc homeostasis via regulating Zap1, Zrt1, and Zrc1 expression [85].

## 4. Detoxification Mechanism of Trace Elements in Yeast

Trace elements become toxic when the concentration of trace elements exceeds the tolerance threshold of yeast cells, and the toxicity is specific. The toxic microelements destroy plasma membrane, bind in a non-specific manner with biomolecules, and interfere with the homeostasis of base metals by competing with their normal transport and buffering systems, resulting in toxic effects which impede yeast growth and metabolism [86]. The toxicity of microelements in cells can cause oxidative stress [87], base damage [88], and altered DNA repair [89], as well as inhibiting enzyme function and interfering with proliferation [90]. Additionally, the cell cycle process is also affected [91], and apoptosis or differentiation of protein function is also impaired [92]. In response to microelements toxicity, yeast cells block cell cycle progression, altering gene expression and metabolism, and tweaking transport processes to protect cell and gene integrity [93]. We summarize essentiality and toxicity values of trace elements in the human body (Table 1).

In the case of metallic chromium, tetravalent chromium is considered the most toxic form. Compared to the tetravalent form, the toxicity of trivalent chromium is very low, and the rapid reduction of tetravalent chromium by intracellular reducing agents (such as ascorbic acid, cysteine, glutathione) to trivalent chromium produces reactive oxygen species (ROS) and chromium in the pentavalent and hexavalent states [94]. ROS combines with proteins, hydrocarbons, and nucleic acids to attack and destroy macromolecules in cells, resulting in protein oxidation, lipid peroxidation, and DNA damage, especially base oxidation and single-chain breakage [93]. Long-term exposure to chromium in its hexavalent state is associated with an increased risk of lung cancer. Throughout the long evolutionary process, organisms must strictly maintain the homeostasis and balance of intracellular metal trace elements in order to adapt to the changing environment.

### 4.1. Mechanism of Glutathione Detoxification

Glutathione is the most abundant non-protein thiol tripeptide, which mainly exists in eukaryotic cells. In cells, glutathione exists mainly in either reduced (GSH) or oxidized (GSSG) form/state [95]. Moreover, glutathione is a crucial redox buffer, reducing the adverse effects of oxidative stress, protecting the mitochondrial macromolecules from the harmful effects of ROS [96], and participating in DNA repair [97].

In *S. cerevisiae*, the enzymes (glutamylcysteine synthase (GSH1) and glutathione synthase (GSH2)) catalyze the continuous glutathione synthesis from three precursor amino acids molecules, viz., glutamate, cysteine, and glycine [98,99]. The thiol group (-SH) in glutathione binds to heavy metals so that heavy metals cannot disrupt cellular metabolic activities [100]. Glutathione is a protein containing cysteine, and cysteine residues can bind metals to overcome the toxicity caused by metals [101]. For selenium, glutathione can not only bind to selenium through cysteine residues, but also is closely related to the generation of nano-selenium. In *S. cerevisiae*, tetravalent selenium ions are easily spontaneously reacted with reduced glutathione to form selenide glutathione (GS-Se-SG) and GSSG [102]. Among them, selenide glutathione was converted to glutathione [103]. Glutathione is catalyzed by superoxide dismutase to convert into elemental selenium (Se^0^), This pathway is widely believed to be the main way for *S. cerevisiae* to reduce selenite to form biological elemental nano-selenium [104].

Thorsen et al. [105] studied the mechanism of arsenic resistance in *S. cerevisiae* and observed that the glutathione biosynthesis pathway was induced by exposure of yeast to arsenic. They also identified the core transcriptional regulators as Yap1p and Met4p, which regulate glutathione synthesis. Lafaye et al. [106] proved that the high production of glutathione in yeast was essential for the detoxification of heavy metals such as cadmium.

### 4.2. Vacuolar Metal Transporter

Yeast and other organisms detoxify heavy metals by chelating the metal with a ligand. There are heavy metal transporters on the vacuole, which transport the metal–protein bound complexes to the vacuole in order to block heavy metals and prevent them from damaging the normal physiological function of the cell. Metal ions enter the organelles such as vacuoles, mitochondria, endoplasmic reticulum, or Golgi apparatus through highly active membrane transporters that isolate metal ion chelates and prevent them from exerting toxic effects [107]. The CCC1 transport system for iron and manganese ions affects the accumulation of iron and manganese ions in vacuoles. It involves the transfer of iron from the cytosol to the vacuole. The overexpression of CCC1 leads to decreased iron content in the cytosol and increased iron content in vacuoles. In contrast, the loss of CCC1 leads to a decrease in iron content and iron storage in vacuoles, thereby affecting the level of iron concentration in the cytosol and cell growth [108]. Yeast cadmium factor 1 (Ycf1) sequesters heavy metals and glutathione into the vacuole to combat cell stress. Ycf1 performs this protective function when conjugated with toxic heavy metals such as cadmium, mercury, or lead into vacuoles [109]. In addition, YCF1 oxidizes glutathione to maintain cellular redox homeostasis and reduce metal-induced cell damage.

### 4.3. Metallothionein and Phytochelin

In addition to glutathione, certain other proteins also chelate with heavy metals for detoxification purposes, including small metal-binding peptides known as phytochelatins (c-Glu–Cys)nGly) and metallothioneins (small cysteine-rich proteins). These metal-chelating proteins neutralize the toxic effect of free metal ions by binding with them inside the cells.

Phytochelatins (PCs) and the vacuolar membrane transporter (SpHMT1) play an important role in the tolerance of heavy metal stress in clonal yeast. PCs are synthesized from glutathione in yeast cells through plant chelate peptide synthase. PCs, as ligands of intracellular heavy metals, form a low molecular weight complex in the cytoplasm, which can enter vacuoles through SpHMT1 and further form stable (phytochelatins–membrane transporter) PC-metal high molecular weight complexes. This sequential process is considered to be the main mode of vacuolar compartmentalization of heavy metals and the main mechanism for heavy metal tolerance among higher plants and fungi [110]. In *S. cerevisiae*, only PC2 ((γ-glutamylcysteinyl)_2_-glycine) can be synthesized by vacuolar serine carboxypeptidase CPC and CPY, which transport the high molecular weight complex to the vacuoles mainly through vacuolar membrane protein YCF1 to achieve tolerance to heavy metals. In a study by Matthias et al. [94], glutathione was found to affect zinc homeostasis, with an increased phytochelatins concentration being related to lower free zinc levels in vacuoles, indicating that phytochelatins are important for zinc buffering in *S. cerevisiae*.

Metallothioneins are a kind of metal-binding protein characterized by more cystine residues and low molecular weight. Metallothionein is a universal conjugate against metal cytotoxicity, which prevents the free metal ions from destroying the cells through chelation [111]. It has the capacity to buffer intracellular metal. The amino and carboxyl groups of metallothionein can bind to metal cations [101]. In addition, the high binding capacity of metallothionein is that it is rich in thiol groups. The thiol group has higher binding ability to metal than the amino group and the carboxyl group [112]. CUP1 is a metallothionein found in *S.*
*cerevisiae*, which has been shown to protect yeast cells against oxidants [113]. Ruta et al. [114] have successfully carried out the heterogenically expression of the metallothionein gene from *Arabidopsis thaliana* and the metallothionein gene of *Thlaspi Caerulescens* in *S. cerevisiae* cells, which increased the resistance of *S. cerevisiae* towards Cu^2+^, Zn^2+^, Cd^2+^, and other heavy metals. In vitro and in vivo studies have shown that low initial doses of cadmium are protective against subsequent high doses of cadmium. This is because cadmium induces the synthesis and storage of metallothioneins in the liver and kidneys [115].

In general, the detoxification mechanisms of yeast for heavy metals include expression of related genes (GSH1 and GSH2, etc.) and proteins (glutathione, etc.) to alleviate oxidative stress [116], transport of heavy metals out of the cell (PCA1, etc.) or into the vacuole by related transporters (YCF1 and BPT1, etc.) [117], and bind metallothionein [118] and phytochelatins [119] with heavy metal ions to reduce their toxicity and adverse effects (Figure 1). The following table summarizes the metal detoxification mechanism and related genes (Table 2). 

## 5. Five Yeasts Enriched with Trace Elements Obtained by Metabolic Engineering

Recently, there have been a number of studies focused on the enrichment of heavy metals in yeast, including optimization of the fermentation process and culture medium system, mutation, and other methods to obtain yeast tolerant to heavy metals, but still, the metal enrichment of yeast has not been greatly improved. As an example, consider chromium-enriched yeast. Yeast enriches chromium by about 1 g/kg, and the highest amount only reaches 4 g/kg [41,124,125]. Several chromium-enriched yeasts only adsorb chromium on the surface of their cells or, more simply, a mixture of yeast and chromium. Until inorganic chromium is converted into active chromium in order to supplement the chromium source, the utilization rate is not improved. The low concentration of metal trace elements in yeast causes a lot of wastewaters containing heavy metals to be produced during yeast production, which is not friendly to the environment. Therefore, many studies have been dedicated to improving the metal enrichment of yeast. Metabolic modification is a possibility, in addition to traditional methods, which is expected to greatly improve metal enrichment. Previous studies on genes related to metal enrichment in yeast have found that the transport and detoxification mechanisms of yeast are intimately related to metal enrichment. Zhao et al. [126] found that 108 yeast single-gene deletion mutants were found to be sensitive to ZnCl_2_ through genome-wide screening, of which 64 mutants had higher intracellular zinc content than the wild type when the concentration of zinc in the medium is high. Berrak et al. [127] obtained yeast mutants that were resistant to metals such as Fe, Zn, Cr, and Co through a reverse metabolic engineering strategy by chemical mutation and iron stress application and found that the expression of phosphate transporter gene PHO84 and iron transporter FTR1 in mutants was down-regulated, and the expression of related genes which are dealing with oxidative stress was found to be up-regulated.

The metal transport and detoxification mechanisms of yeast have been described in detail above. Considering their close relationship and the enrichment of metal trace elements, strategies for metabolic engineering of yeast may also include overexpression of membrane transporter activity, overexpression of metal-isolating organelle transporters (e.g., vacuolar transporters), deletion of export transporter genes, and enhancement of detoxification pathways to enhance metal tolerance by yeast [122].

Overexpression of the membrane transport can enhance the transport of microelement ions into yeast cells. More trace elements enter yeast cells and stimulate them to produce more glutathione and other proteins. Activated metal protein chelates are formed when these proteins combine with metal trace elements [128]. In order to prevent the excess of metal trace elements and metal toxicity in yeast cells, it is extremely crucial to enhance the detoxification pathway of yeast and the synthesis of metal-chelating high-affinity ligands, such as glutathione pathway, metallothionein, and PC as mentioned above [63,93]. Metal chelators not only alleviate the cytotoxicity caused by excessive metal but also chelate with metal and reduce the adverse effects caused by them. Finally, the organelle transporters on the surface of vacuoles, Golgi apparatus, and endoplasmic reticulum membrane were overexpressed in order to transport metals and protein chelates into organelles for isolation [108].To improve gene expression, not only can key metabolic pathway genes be overexpressed, but multiple copies of genes can also be constructed. In yeast, multicopy expression of genes may increase the transport of metal trace elements, although it is also possible that yeast with a high copy number of genes has low metal enrichment capability.

Many studies have carried out metabolic engineering on transporters (Table 3). Overexpression of the sulphate osmotic enzymes Sul1 and Sul2 shows a more than five-fold increase in metal absorption of chromate [122]. The overexpression of ZRT1 and ZRT2 transporters greatly improved the transport capacity of Zn by about 10-fold. On the other hand, overexpression of CTR1 and CTR3 greatly promoted copper transport by about 10-fold [122]. SMF1 has a wide range of metal specificity and is often optimized and engineered to enhance the absorption of metals including Fe and Ni [128]. Express hyperactive Aft1 alleles Aft1-1up lead to iron cells accumulating up to four-fold more endogenous iron than Aft1-expressing cells [79]. The co-expression of cell membrane transporter (SMF1) and CCC1 increased the uptake of Mn and Cd more than 10-fold [122].

Enhancing the detoxification mechanism of glutathione is an important method of increasing the metal tolerance in yeast. Many studies have explored the glutathione pathway and discussed the methods to effectively enhance this pathway. Perrone et al. [132] reported that a number of genes (PEP12, UBP6, etc.) are involved in the production of glutathione. Previously, Hara et al. [103] cultivated an engineered strain of *S. cerevisiae* whose glutathione biosynthesis was boosted by overexpression of MET16, which showed a content of intracellular glutathione 1.5 times higher than that of the parent strain. Many studies have been reported where maximizing glutathione production was discussed. These mainly involved two promising approaches: increasing the biomass concentration of glutathione producing yeast and increasing the intracellular glutathione content [98]. Aside from chelating metal itself, glutathione can also produce metallothionein and phytochelatin to further chelate metal detoxification. Although the content of proteins related to detoxification such as glutathione can be increased through metabolic engineering, people rarely use their enhancement to achieve higher trace element enrichment in *S. cerevisiae*. The detoxification metabolism of many microelements is similar, and some transporters have the same recognition and transport functions for multiple microelement ions or microelement ions of the same valence state. *S. cerevisiae* is a model eukaryote. The technical tools (CRISPR/Cas9 [133], etc.) and expression cassette elements (promoters [134] and terminators, etc., [135]) necessary for metabolic engineering transformation have been fully reported and studied, and the metabolic engineering transformation of *S. cerevisiae* has been successfully carried out. In industry, engineered *S. cerevisiae* has been successfully applied in the production of terpenes [136] and fatty acid derivatives [137]. As a result, we can rearrange yeast-related genes or integrate foreign genes through metabolic engineering. As a result, a more efficient production strategy for trace element nutritional supplements might be applied with reduced production period and pollution costs.

## 6. Conclusions and Future Perspectives

A lack of trace elements can lead to a variety of health problems. Trace elements, such as chromium, are difficult to ingest from food, so nutritional supplements are needed to assist ingestion. More and more metal trace element nutritional supplements are needed to improve the deficiency of metal trace elements in the diet. The advantages of yeast-enriched trace elements include high efficiency, green, stability, and extensive nutrition, thus making them ideal for supplementing trace elements in humans.

Combined with the previous description of microelements transport and detoxification mechanism, we have reason to believe that *S. cerevisiae* can be engineered to hyper accumulate metals efficiently by overexpression of certain transporter genes and evolving native metal transporters and engineering mechanisms for metal detoxification. Engineered yeasts have more applications beyond nutritional supplements because of their enhanced ability to enrich metals. Yeast is also a common raw material for making bread, beer, and many other foods. To make food rich in trace elements, yeast rich in trace elements can be substituted for ordinary yeast. Meanwhile, the metallic taste of the micronutrient supplement can also be improved by the flavor of the food. Furthermore, adding microelement-enriched yeast to feed can improve the immunity of poultry and livestock, reducing the use of antibiotics. Similarly, metabolized yeast can be applied to soil or wastewater contaminated by heavy metals.

Nevertheless, an accumulation of trace metal elements among consumers from metabolized yeast still poses food safety issues. Since metal-enriched yeast food supplements are not authorized, more information is required for their bioavailability. Furthermore, what the metal active substances in the yeast enriched with metal trace elements are and whether these substances can be isolated from yeast to discover a new method to detoxify the yeast remain unclear. As an alternative to the overexpression of import transporters, the elimination of export transporters can also improve metal retention in yeast. However, at present, the export transport mechanism of yeast is not well understood, and research is scarce. This may be a new research direction and a means to further enhance metal enrichment in yeast. Metal and protein chelate are transported by a transporter on the organelle membrane. It is possible that this transporter is localized in the inner membrane of the cell for expression so that yeast can reverse transport active trace elements throughout the body, which may help solve the current safety risk for yeast as a nutritional supplement.

At the same time, whether new forms of trace element supplements, such as nano-chromium and nano-iron and the evaluation of their safety and the comparison of their absorption and utilization efficiency can be obtained, will become the future exploration direction.

As a result of the low enrichment of metal trace elements by yeast, even the yeast with the highest enrichment at present still has a low metal conversion rate in the culture medium, and it requires a certain high concentration environment to achieve such enrichment. The result will be a large amount of waste raw materials and waste liquids containing heavy metals. In addition to the above strategy, yeast can also be used to improve the utilization of raw metal. Metals with low-affinity transport pathways can absorb metals in environments with low metal concentrations. In the future, the low-affinity transport pathway of yeast can be metabolically modified to make yeast achieve high enrichment and high conversion rates at low metal concentrations. In addition to saving raw materials, this method can be more environmentally friendly.

## Figures and Tables

**Figure 1 microorganisms-10-01746-f001:**
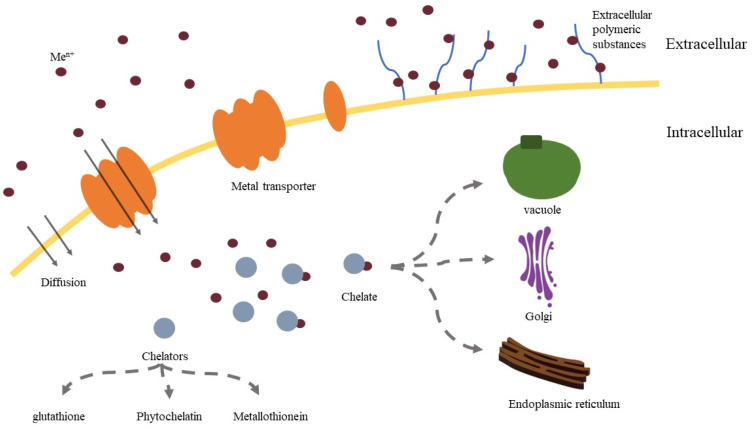
Model for transport and detoxification mechanism of metal trace elements by yeast.

**Table 1 microorganisms-10-01746-t001:** Summary of essentiality and toxicity data for metal trace elements.

Element	IOM FNB		WHO	Labor Ministry DRIs				FDA	SCF		CNS		
	RDA (mg/d)	UL (mg/d)	PMTDI Upper Limit (mg/d)	RDA (Mg/D) 18–30 Years		UL (mg/d)		RDI (mg/d)	DRV (mg/d)		RDA (mg/d) 18–50 Years		UL(mg/d) 18–50 Years
				M	F	M	F		M	F	M	F	
Iron	8 (M and F > 51 years)	45	56 (0.8 mg/kg-day)	7	6	50	40	18	NA	NA	18	20	40
Zinc	11(M)8(F)	40	NA	12	9	40	35	15	9.5	7	12.5	7.5	40
Chromium	NA	NA	NA	0.04	0.03	NA	NA	0.12	NA	NA	0.03	0.03	NA
Copper	0.9	10	35 (0.5 mg/kg-day)	0.9	0.7	10	10	2	1.1	NA	0.8	0.8	8
Selenium	0.055	0.4	NA	0.03	0.025	0.28	0.22	0.07	0.055	NA	0.06	0.06	0.4
Molybdenum	0.045	2	NA	0.025	0.02	0.55	0.45	0.075	NA	NA	0.1	0.1	0.9
Manganese	NA	11	NA	4.0	3.5	11	11	2	1–10	NA	4.5	4.5	11

mg/d: mg/day; NA, not available. IOM FNB: the U.S. Food and Nutrition Board of the Institute of Medicine; WHO: World Health Organization; FDA: Food and Drug Administration; SCF: Scientific Committee of Food; CNS: Chinese Nutrition Society; RDA: recommended dietary allowance; UL: tolerable upper intake level; PMTDI: permissible maximum tolerable daily intakes; RDI: reference daily intake; DRV: dietary reference values; the values provided are for both males and females, unless (M) for males and (F) for females is noted.

**Table 2 microorganisms-10-01746-t002:** Resistance mechanisms of yeast towards heavy metal ions.

Resistance Mechanisms	Related Gene	Functional Molecule	Function	Reference
Extracellular sequestration		Extracellular polymeric substances	Efficient biosorbents to various heavy metals	[101,120]
	PCA1	Heavy metal transporters	cadmium efflux pump	
Glutathione detoxification	GSH1	Glutamyl cysteine synthase	Prevents oxidation or irreversible binding of functional proteins to metals	[121]
	GSH2	Glutathione synthase		
Intracellular sequestration	CCC1	Vacuolar transporter	Transport the metal into the vacuole	[109,110,122]
	COT1			
	SpHMT1			
	YCF1	Yeast cadmium factor		
Phytochelatins detoxification	PRC1	Serine carboxypeptidase; Synthesize PC2	Metal-chelator	[123]
	CPC CPY			
Metallothioneins detoxification	Cup1	Metallothionein	Metal-chelator	[113]

**Table 3 microorganisms-10-01746-t003:** Metabolic engineering to enrich trace elements by *S. cerevisiae*.

Host Strain	Description	Enriched Elements	Multiple of Increase (Compared to Host Strain)	Reference
W303α	*Zrt1Δ:: PGAL1-* *Zrt1; Zrt2Δ:: PGAL1-* *Zrt2*	Zn	10-fold	[122]
	*FET4Δ:: PGAL1-* *FET4; SMF1Δ:: PGAL1-* *SMF1*	Fe, Zn, Cu, Mn	3 to 5-fold	
	*Sul11Δ:: PGAL1-* *Sul1; Sul22Δ:: PGAL1-* *Sul22*	Cr	5-fold	
BY4711	*AFT1Δ:: AFT1-UP*	Fe	4-fold	[79]
BY4711	*PHO84Δ:: PGAL1-* *PHO84*	Mn, Co, Cu	1.7 to 2.8-fold	[129]
CEN.PK113-7D	*leu2:: YIpOptSMT, YCT-CHI*	Se	8-fold	[130]
BY4743	*CCC1Δ:: PTEF2-CCC1*	Fe	8-fold	[131]

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
