# Peer review of "Accumulation and Enrichment of Trace Elements by Yeast Cells and Their Applications: A Critical Review"

_microorganisms, 2022, doi:10.3390/microorganisms10091746_

Round 1
Reviewer 1 Report
This review compares popular forms of trace metal supplementation. The article analyzes the mechanism of detoxification and transport of trace elements of heavy metals in yeast. In addition, it summarized the genes involved in the transport and detoxification of trace elements, and the construction of engineered yeast strains enriched with trace elements through metabolic modification was analyzed.
The topic is interesting and, in the opinion of the reviewer, may attract a sufficiently wide audience. The authors are experienced in this area, and the overall organization of the manuscript seems appropriate. The literature review is good and up to date.
Nevertheless, throughout the text, there are formatting errors that need to be corrected, such as the following:
· Abstract – line 13-14 - please redraft to remove repetitions „enriched yeast”
· Table 1 - please change the title of table 1
· Table 1 - please explain the units [mg/d] used. Whether it is a daily dose?
· Table 1 - why the units in the table are not standardized? The headings are given in mg/d, while inside the table many data is given in µg/d. Please sort it out / standardize it.
· Line 134 „ferrous diglycine” rather „ferrous diglicinate” – please correct
· Line 192 – „Maria et al. [33]” - discrepancy in citation not in accordance with the literature references
· Line 394-395 whether the units of mg/kg are appropriate? If so please express 1000 or 4000 mg in g/kg
· Please check the text again, spaces are missing in many places – line 147, 192, 194, 204, 206, 207, 213 etc…
· The references chapter should be checked carefully
· I also recommend submitting your article to the editorial service in English before resubmitting
Author Response
Thank you very much for your suggestions. Here is my response.
This review compares popular forms of trace metal supplementation. The article analyzes the mechanism of detoxification and transport of trace elements of heavy metals in yeast. In addition, it summarized the genes involved in the transport and detoxification of trace elements, and the construction of engineered yeast strains enriched with trace elements through metabolic modification was analyzed.
The topic is interesting and, in the opinion of the reviewer, may attract a sufficiently wide audience. The authors are experienced in this area, and the overall organization of the manuscript seems appropriate. The literature review is good and up to date.
Nevertheless, throughout the text, there are formatting errors that need to be corrected, such as the following:
Response:
Many thanks for your constructive suggestions. According to your suggestions, I made the following modifications to my manuscript.
Abstract – line 13-14 - please redraft to remove repetitions „enriched yeast”
Response:
I have rewritten the sentence. Thank you.
Table 1 - please change the title of table 1
Response:
I have changed the title. Thank you.
Table 1 - please explain the units [mg/d] used. Whether it is a daily dose?
Response:
mg/d means mg/day. And I have added the explanation under the table. Thank you.
Table 1 - why the units in the table are not standardized? The headings are given in mg/d, while inside the table many data is given in µg/d. Please sort it out / standardize it.
Response:
I have standardized the units. Thank you.
Line 134 „ferrous diglycine” rather „ferrous diglicinate” – please correct
Response:
I have corrected. Thank you.
Line 192 – „Maria et al. [33]” - discrepancy in citation not in accordance with the literature references
Response:
I have corrected the reference. Thank you.
Line 394-395 whether the units of mg/kg are appropriate? If so please express 1000 or 4000 mg in g/kg
Response:
Thank you for your advice. I have corrected.
Please check the text again, spaces are missing in many places – line 147, 192, 194, 204, 206, 207, 213 etc…
Response:
I have checked and corrected. Thank you.
The references chapter should be checked carefully
I also recommend submitting your article to the editorial service in English before resubmitting
Response:
I have checked and corrected. Thank you.

Reviewer 2 Report
This article provides an overview of the status of research on metal uptake and accumulation in the yeast S. cerevisiae.
It states the facts from the viewpoint of applying such yeasts for nutritional purposes.
The authors focus their review on: (1) different ways to supplement, (2) trace elemnt transport in yeast, (3) detoxification mechanisms in yeast, (4) enrichment by metabolic engineering.
This focus misses out the aspect of traditional (fermentation) methods to enrich metal trace elements. I suggest to include a chapter on this aspect (compare line 391), as it is probably still the most widely used method.
And including data on this aspect would also better allow the evaluation of the metabolic engineering approach. In the current state of the manuscript the authors merely claim that the genetic engineering approach is the most promising. However, data are not presented to prove this.
An additional table is recommended that gives an overview on processes and metabolic engineering approaches.
Some questions, comments and recommendations along the manuscript:
Numbering: Chapter 3 is missing
Abstract: line 15: “metal trace element enriched yeast is more efficient….”: please present data on this statement in the text. If this is not referenced, the abstract should rather not mention this statement.
Abstract: line 16/17: “engineered yeast can enrich the high concentration…” I cannot find a clear reference on this statement in the text. Please modify or improve the text accordingly.
Line 21: “engineering” should probably read “engineered”
Line 97 / 98: please provide a reference
Chapter 4.2: please include the work of Ramos-Alonso 2018 (doi: 10.1039/c8mt00124c)
Chapter 4.3: please include the work of Bian et al. 2021 (doi: 10.1016/j.jtemb.2021.126759)
Table 2: this table would better fit in Chapter 5
Chapter 6: I recommend to include a table to give a better overview on genes, pathways and especially on results obtained.
Line 423: “optimization of the membrane” should probably read: “optimization of the membrane transport”.
Line 450/1: the paper of Hara et al describes the modification of glutathione biosynthesis. How does this change trace metal elements accumulation? Please edit.
Copper: there are a number of papers on copper transport and copper enrichment. These were not included. Please comment.
Line 459-461: this conclusion would be better substantiated if amounts / fold increases for accumulation were precisely referenced. The last sentence says “can be developed”: is this meant to say: yes “it has been possible” or rather “might be possible in the future”?
Line 469: also in the Conclusion section: ‘”yeast can be engineered”: please clearly demonstrate this in Chapter 6. Line 500 refers to “in the future”: It would help the impact of the review very much if the authors were explicit what is achieved today and is feasible in the future.
Author Response
Thank you very much for your suggestions. Here is my response.
This article provides an overview of the status of research on metal uptake and accumulation in the yeast S. cerevisiae.
It states the facts from the viewpoint of applying such yeasts for nutritional purposes.
The authors focus their review on: (1) different ways to supplement, (2) trace elemnt transport in yeast, (3) detoxification mechanisms in yeast, (4) enrichment by metabolic engineering.
This focus misses out the aspect of traditional (fermentation) methods to enrich metal trace elements. I suggest to include a chapter on this aspect (compare line 391), as it is probably still the most widely used method. And including data on this aspect would also better allow the evaluation of the metabolic engineering approach.
Response:
Thank you very much for your constructive suggestions. I have added relevant contents of traditional fermentation methods in Section 2.3 of the manuscript (Line 193 to 205). In addition, section 2.3.1 to 2.3.4 (Line 227 to 234, 242 to 247, 254-263, 287 to 290) lists the production data of fermentation method.
In the current state of the manuscript the authors merely claim that the genetic engineering approach is the most promising. However, data are not presented to prove this. An additional table is recommended that gives an overview on processes and metabolic engineering approaches.
Response:
Many thanks for your constructive suggestions. According to your suggestion, in Chapter 5 of the manuscript, I added the table of metabolic engineering to enrich trace elements by S. cerevisiae. Meanwhile, I summarized the relevant genes in Table 2. In the chapter 5 of the manuscript, I listed the possible methods and approaches of metabolic engineering to transform S. cerevisiae to obtain trace element enriched yeast (Line 559 to 600). Indeed, there have been articles that increase the intracellular metal of yeast and enhance the detoxification pathway through metabolic engineering. However, few people obtain trace element-enriched yeast through metabolic engineering. Thank you.
Some questions, comments and recommendations along the manuscript:
Numbering: Chapter 3 is missing
Abstract: line 15: “metal trace element enriched yeast is more efficient….”: please present data on this statement in the text. If this is not referenced, the abstract should rather not mention this statement.
Abstract: line 16/17: “engineered yeast can enrich the high concentration…” I cannot find a clear reference on this statement in the text. Please modify or improve the text accordingly.
Line 21: “engineering” should probably read “engineered”
Response:
I have corrected. Thank you.
Line 97 / 98: please provide a reference
Response:
References have been added. Thank you.
Chapter 4.2: please include the work of Ramos-Alonso 2018 (doi: 10.1039/c8mt00124c)
Chapter 4.3: please include the work of Bian et al. 2021 (doi: 10.1016/j.jtemb.2021.126759)
Response:
I have added. Thank you.
Table 2: this table would better fit in Chapter 5
Response:
I have modified. Thank you.
Chapter 6: I recommend to include a table to give a better overview on genes, pathways and especially on results obtained.
Response:
Many thanks for your constructive suggestions. According to your suggestion, in Chapter 5 of the manuscript, I added the table of metabolic engineering to enrich trace elements by S. cerevisiae. Meanwhile, I summarized the relevant genes in Table 2, the pathways of metabolic engineering and the results obtained in Chapter 5 (Line 559 to 600).
Line 423: “optimization of the membrane” should probably read: “optimization of the membrane transport”.
Response:
I have corrected. Thank you.
Line 450/1: the paper of Hara et al describes the modification of glutathione biosynthesis. How does this change trace metal elements accumulation? Please edit.
Response:
I have added the explanation (Line 583 to 587) in manuscript. Thank you.
Copper: there are a number of papers on copper transport and copper enrichment. These were not included. Please comment.
Response:
Many thanks for your constructive suggestions. The enrichment and transport of copper have been described in this paper, such as line 76, 154, 164, 246. However, the human body rarely lacks copper, so copper supplements are not specifically described in the manuscript. Thank you.
Line 459-461: this conclusion would be better substantiated if amounts / fold increases for accumulation were precisely referenced. The last sentence says “can be developed”: is this meant to say: yes “it has been possible” or rather “might be possible in the future”?
Response:
I have modified. Thank you. The S. cerevisiae transformed by metabolic engineering has not been produced on a large scale, so it’s “might be possible in the future”.
Line 469: also in the Conclusion section: “yeast can be engineered”: please clearly demonstrate this in Chapter 6. Line 500 refers to “in the future”: It would help the impact of the review very much if the authors were explicit what is achieved today and is feasible in the future.
Response:
Your good advice was very much appreciated. According to your suggestion, I added the statement that S. cerevisiae can be transformed by metabolic engineering in Chapter 5 (Line 592 to 598) to increase the persuasiveness of the conclusion.

Reviewer 3 Report
The article is not well written. Abstract is very badly described, nothing comes of it, there are no details, no prospects for the future. The authors very poorly present a research problem that is not specifically described. Each section requires a thorough improvement and the addition of newer information to enrich the presented manuscript. The article cannot be accepted in its current form. The authors of the publication do not present any detoxification mechanisms that may affect yeast cells under the influence of elements. Each detoxification process should be described. There is no description of individual proteins of great importance, peroxisomes, exosomes, etc. There is no description of yeast biomass as a protein source. This step needs to be expanded and more information added. This will expand the manuscript and will be more interesting for the readers. The manuscript lacks a description of the prospects for using yeast biomass in various industrial sectors, also in the food industry. There is no description of selenium yeast and their specific characteristics. This topic should be specifically presented. Each form of yeast enriched with a given element should be described in terms of advantages and disadvantages as well as its impact on the environment or the human and animal organism. The authors presented the limits of elements that should be present in yeast biomass. The authors did not write anything that when arranging food doses, technologists, breeders should pay attention not only to the appropriate ratio of energy, protein and dry matter, but also take into account the content of micro- and macroelements and other substances present in trace amounts, and often of key importance in the proper functioning of the body. Enriching food with micronutrients with the use of microorganisms in the process of introducing these elements into the food chain is a cheap and safe method, and moreover, more beneficial for the health of consumers from the point of view of ecological prophylaxis than pharmacological supplementation. It needs to be described in more detail and data from various international offices, such as WHO, etc., should be added. There is a lack of information on the production of element-enriched yeast. For the production of e.g. feed yeast is used a strain of Saccharomyces cervisiae, which is grown on a suitable medium (molasses or gluten) in the presence of salts of various elements, e.g. inorganic forms. This allows the inorganic matter to be optimally incorporated into the cells. During the fermentation process, yeast takes up elements and incorporates them in the form of various compounds into the body's own structures. About the different forms of elements in the cell cytosol must be described in the text.
Author Response
Thank you very much for your suggestions. Here is my response.
The article is not well written. Abstract is very badly described, nothing comes of it, there are no details, no prospects for the future. The authors very poorly present a research problem that is not specifically described. Each section requires a thorough improvement and the addition of newer information to enrich the presented manuscript. The article cannot be accepted in its current form.
Response:
Many thanks for your constructive suggestions. The abstract part of the manuscript was modified and the contents of each chapter were expanded.
The authors of the publication do not present any detoxification mechanisms that may affect yeast cells under the influence of elements. Each detoxification process should be described. There is no description of individual proteins of great importance, peroxisomes, exosomes, etc.
Response:
Your good advice was very much appreciated. The description of glutathione and metallothionein detoxification functional group has been added in sections 4.1(Line 431 to 441) and 4.3(Line 489 to 492). In Chapter 4 of the manuscript, I described the detoxification mechanism of yeast, including the antioxidant mechanism of glutathione, transport of heavy metals out of the cell or into the vacuole by related transporters, binding of metallothionein and phytochelatins with heavy metal ions to reduce their toxicity and adverse effects. And the detoxification pathway and functional molecule of Saccharomyces cerevisiae are summarized in Table 2. Thank you.
There is no description of yeast biomass as a protein source. This step needs to be expanded and more information added. This will expand the manuscript and will be more interesting for the readers. The manuscript lacks a description of the prospects for using yeast biomass in various industrial sectors, also in the food industry.
Response:
Many thanks for your constructive suggestions. The protein in yeast biomass is very important for the enrichment of trace elements. Yeast protein may become an important source of protein in the future. I added relevant contents in Section 2.3 (Line 180 to 190) of the manuscript.
There is no description of selenium yeast and their specific characteristics. This topic should be specifically presented. Each form of yeast enriched with a given element should be described in terms of advantages and disadvantages as well as its impact on the environment or the human and animal organism. The authors presented the limits of elements that should be present in yeast biomass.
Response:
Thank you very much for your comments. Selenium yeast is an important part of trace element enriched yeast and also an important reference for S. cerevisiae to enrich metal trace elements. Your suggestion is very critical. Selenium-enriched yeast has been added in section 2.1 (Line 148 to 150), section 2.2 (Line 172 to 176), section 2.3.4 (Line 274 to 290), section 3.1 (Line 341 to 345), section 4.1 (Line 432 to 441), section 6 (Line 641 to 645) of the manuscript. In Section 2.3.1 (Line 218 to 226) , 2.3.2 (Line 240 to 243), 2.3.3 (Line 251 to 254), 2.3.4 (Line 278 to 281)of the manuscript, the advantages of trace element enriched yeast and benefits to human body are described. Its contribution to environment is in Chapter 5 (Line 520 to 522) and Chapter 6 (Line 653 to 658)
The authors did not write anything that when arranging food doses, technologists, breeders should pay attention not only to the appropriate ratio of energy, protein and dry matter, but also take into account the content of micro- and macroelements and other substances present in trace amounts, and often of key importance in the proper functioning of the body. Enriching food with micronutrients with the use of microorganisms in the process of introducing these elements into the food chain is a cheap and safe method, and moreover, more beneficial for the health of consumers from the point of view of ecological prophylaxis than pharmacological supplementation. It needs to be described in more detail and data from various international offices, such as WHO, etc., should be added.
Response:
Many thanks for your constructive suggestions. In Table 1 of the manuscript, the tolerable upper intake levels of several metal trace elements are summarized. Data from different organizations (WHO, etc.) are also provided in Table 1. Regarding the appropriate proportion of energy, protein and dry matter, yeast enriched in trace elements can indeed provide nutrition other than trace elements. For example, as protein source, I added the relevant content of yeast protein in Section 2.3 (Line 180 to 190). I think how to supplement nutrition and how to adjust the optimal proportion will also be important research directions in the future.
There is a lack of information on the production of element-enriched yeast. For the production of e.g. feed yeast is used a strain of Saccharomyces cervisiae, which is grown on a suitable medium (molasses or gluten) in the presence of salts of various elements, e.g. inorganic forms. This allows the inorganic matter to be optimally incorporated into the cells. During the fermentation process, yeast takes up elements and incorporates them in the form of various compounds into the body's own structures. About the different forms of elements in the cell cytosol must be described in the text.
Response:
Many thanks for your constructive suggestions. We looked for the content of trace elements in the medium components. The content of trace elements in molasses is very small or even undetectable, which is far lower than the inorganic salt added in the production of trace element enriched yeast. The effect on fermentation is very limited. Thank you again for your suggestions.

Round 2
Reviewer 3 Report
The Articleai is better organizer. I accept this manuscript.